# Implementing HLA-B*58:01 testing prior to allopurinol initiation in Malaysian primary care setting: A qualitative study from doctors' and patients' perspective

**Wei Leik Ng**[1]*, **Norita Hussein**[1]*, **Chirk Jenn Ng**[1,2,3], **Nadeem Qureshi**[4], **Yew Kong Lee**[1], **Zhenli Kwan**[5], **Boon Pin Kee**[6‡], **Sue-Mian Then**[7‡], **Tun Firzara Abdul Malik**[1‡], **Fatimah Zahrah Mohd Zaidan**[8‡], **Siti Umi Fairuz Azmi**[9‡]

1 Department of Primary Care Medicine, Faculty of Medicine, Universiti Malaya, Kuala Lumpur, Malaysia, 2 Department of Research, SingHealth Polyclinics, Singapore, Singapore, 3 Health Services and Systems Research, Duke-NUS Medical School, Singapore, Singapore, 4 Division of Primary Care, School of Medicine, University of Nottingham, Nottingham, United Kingdom, 5 Division of Dermatology, Department of Medicine, Faculty of Medicine, Universiti Malaya, Kuala Lumpur, Malaysia, 6 Department of Biomedical Science, Faculty of Medicine, Universiti Malaya, Kuala Lumpur, Malaysia, 7 Division of Biomedical Science, School of Pharmacy, Faculty of Science and Engineering, University of Nottingham, Selangor, Malaysia, 8 Tanglin Health Clinic, Ministry of Health Malaysia, Kuala Lumpur, Malaysia, 9 Taman Ehsan Health Clinic, Ministry of Health Malaysia, Selangor, Malaysia

☯ These authors contributed equally to this work.
‡ BPK, SMT, TFAM, FZMZ and SUFA also contributed equally to this work.
* wlng@ummc.edu.my (WLN); norita@ummc.edu.my (NH)

**Data Availability Statement:** The data that support the findings of this study belongs to University of Malaya Medical Centre (UMMC) and Ministry of

## Abstract

### Introduction

Allopurinol, the first-line treatment for chronic gout, is a common causative drug for severe cutaneous adverse reactions (SCAR). HLA-B*58:01 allele was strongly associated with allopurinol-induced SCAR in Asian countries such as Taiwan, Japan, Thailand and Malaysia. HLA-B*58:01 screening before allopurinol initiation is conditionally recommended in the Southeast-Asian population, but the uptake of this screening is slow in primary care settings, including Malaysia. This study aimed to explore the views and experiences of primary care doctors and patients with gout on implementing HLA-B*58:01 testing in Malaysia as part of a more extensive study exploring the feasibility of implementing it routinely.

### Methods

This qualitative study used in-depth interviews and focus group discussions to obtain information from patients with gout under follow-up in primary care and doctors who cared for them. Patients and doctors shared their gout management experiences and views on implementing HLA-B*58:01 screening in primary care. Data were coded and analysed using thematic analysis.

### Results

18 patients and 18 doctors from three different healthcare settings (university hospital, public health clinics, private general practitioner clinics) participated. The acceptability to HLA-

Health Malaysia. Restrictions by the ethics committee apply to the availability of these data as the data contain potentially identifying participant information and sensitive information, and so are not publicly available. Data can be requested from the UMMC ethics committee (contact number: +603 7949 3209, email: ummcmrec@ummc.edu. my) and Medical Research and Ethics Committee (MREC) Malaysia (contact number: +603-3362 8398, email: mrecsec@moh.gov.my).

**Funding:** The authors (WLN, NH, CJN, YKL, ZK, BPK, TFAM) disclosed receipt of the following financial support for the research: This work was supported by the University of Malaya Specialist Centre (UMSC) CA.R.E Fund Research Grant 2020 (grant number: PV016-2020, URL: https://resfom. um.edu.my). The funders had no role in study design, data collection and analysis, decision to publish, or preparation of the manuscript.

**Competing interests:** The authors have declared that no competing interests exist.

B*58:01 screening was good among the doctors and patients. We discovered inadequate disclosure of severe side effects of allopurinol by doctors due to concerns about medication refusal by patients, which could potentially be improved by introducing HLA-B*58:01 testing. Barriers to implementation included out-of-pocket costs for patients, the cost-effectiveness of this implementation, lack of established alternative treatment pathway besides allopurinol, counselling burden and concern about genetic data security. Our participants preferred targeted screening for high-risk populations instead of universal screening.

## Conclusion

Implementing HLA-B*58:01 testing in primary care is potentially feasible if a cost-effective, targeted screening policy on high-risk groups can be developed. A clear treatment pathway for patients who test positive should be made available.

## Introduction

Gout is a common disease encountered in primary care with a projected increase in prevalence and incidence worldwide, attributed mainly to changing diet patterns and rising prevalence of metabolic syndrome [1]. Allopurinol is the commonest drug used in treatment for chronic gout, accounting for 80–90% of the prescriptions [2]. However, allopurinol is also one of the most common causative drugs for severe cutaneous adverse reactions (SCAR) [3–6]. In Malaysia, the incidence of allopurinol-induced SCAR was estimated at 2.5 cases per 1000 new allopurinol users, similar to the incidence reported in other Asian countries such as Taiwan and Thailand [7].

In 2005, Hung et al. discovered that HLA-B*58:01 allele was strongly associated with the development of allopurinol-induced SCAR, and this association was replicated in many different Asian populations, such as Japan, Thailand and Malaysia [8–11]. In 2020, American College of Rheumatology had conditionally recommended HLA-B*58:01 testing for selected populations with higher HLA-B*58:01 allele frequency, such as those of Southeast Asian descent (e.g., Han Chinese, Korean, Thai) and for African American patients before starting allopurinol [12]. The Clinical Pharmacogenetics Implementation Consortium, an international consortium interested in facilitating pharmacogenetic test use, also supported HLA-B*58:01 testing, especially in Asian population, to reduce the incidence and risk of allopurinol-induced SCAR [13].

Pharmacogenomics, the study of how genetic differences affect an individual's response to medication, is a growing field with more established guidelines on pharmacogenomic testing being made available [14]. However, the uptake of pharmacogenetic testing such as HLA-B*58:01 in primary care, including Malaysia, has been slow. Previous literature had reported that lack of evidence or guidelines for clinical use, higher cost of testing, limited knowledge of doctors and lack of consultation time were among the barriers doctors faced to implement pharmacogenetic testing in general [15, 16]. For the patients, there were concerns about the cost of testing, confidentiality of the genetic data and potential psychological distress from the result, although they generally thought the testing would be beneficial [15, 17].

HLA-B*58:01 screening is not routinely practised in Malaysia with only limited health centres and private laboratories offering the test. In Malaysia, the healthcare system consists of

government-run universal services and the concomitant private healthcare system. While public healthcare services are heavily subsidised, patients still need to pay out of pocket for tests and medications that are less commonly used and usually more expensive, such as HLA-B*58:01 testing. At the time of writing, HLA-B*58:01 test would cost RM250 (USD 54) in one of the study sites for this research. The median monthly income for employed citizens in Malaysia was RM2400 (USD 518) with an average monthly household income of RM8479 (USD 1831) in 2022 [18].

This qualitative study is part of a larger study exploring the feasibility of implementing routine HLA-B*58:01 testing before initiating allopurinol in the Malaysian primary care setting. As the stakeholders for the test are patients and doctors, this study explored the views and opinions of primary care doctors and patients with gout on implementing HLA-B*58:01 testing in Malaysia.

## Methods

### Study design and duration

This study employed a qualitative methodology, using semi-structured interviews and focus groups, to explore the views and opinions of primary care doctors and patients with gout on implementation of HLA-B*58:01 testing in the primary care setting in Malaysia. A descriptive interpretive approach was used to describe and interpret the collective opinions of the participants in the health setting [19]. These interviews were conducted between 6 October 2020 to 28 February 2022.

### Study sample and recruitment

The participants of this study consisted of primary care doctors and patients with gout from three different health settings: 1) a primary care clinic at a hospital site, University of Malaya Medical Centre (UMMC), 2) two Ministry of Health Malaysia community primary care clinics (also known as health clinics), and 3) private general practitioner (GP) clinics. All these health centres are located in a central urban area in Malaysia with a high density of patients with chronic diseases including gout.

The researchers identified the doctors from UMMC and health clinics, with eligibility as having experience managing gout patients. The doctors were selected purposively based on gender, years of practice and availability of postgraduate qualification for primary care. Snowballing sampling was used to identify private GPs for the interview. The first step in sampling the GPs began with personal contacts by the researchers, and the GPs were asked to identify other GPs who might be interested in participating in the interview.

The researchers identified the patients in UMMC and health clinics from the electronic medical records. The patients were selected purposively based on their age, gender, ethnicity, duration of gout, and whether they were taking allopurinol to achieve maximal variation in terms of their opinions, healthcare experiences and needs. For the patients in private GP settings, the GP participants helped to identify patients with gout attending their clinics for follow-up. Once identified, the patients were screened for eligibility using the following criteria: 1) age 18 years old and above, and 2) had clinical diagnosis of gout by doctors and laboratory evidence of hyperuricaemia (hyperuricaemia is defined as serum uric acid >0.42 mmol/L in males and >0.36 mmol/L in females).

The eligible participants were contacted via phone and recruited by researchers or a research assistant. Appointments were set for the participants to be interviewed face-to-face or online via Zoom. The participants were reimbursed RM50 (approximately USD 12) for their effort and time spent on the interview.

## Data collection and instrument

Separate interview guides for patients and physicians were developed. For patients with gout, the consideration was the decision-making process associated with testing for HLA-B*58:01 allele prior to initiation of allopurinol. Ottawa Decision Support Framework (ODSF) was used to inform development of the topic guide for patients [20]. ODSF identified factors (decisional conflict, sociodemographic profile, social support, etc) that influence patients' decision making.

Theoretical Domain Framework (doctor's factor, organizational factor, etc) was used as the conceptual framework to explore feasibility of implementing HLA-B*58:01 testing from the perspective of primary care doctors [21]. This framework helped explore potential issues that could influence the doctors' decision to implement HLA-B*5801 testing in their clinics.

The participants would go through the participant information sheet and ask questions if any. Written consents were obtained after they had understood and agreed to participate in the study. The consent included audio or video recordings (if using Zoom) of the interviews. Information regarding the risk of SCAR with allopurinol, association of HLA-B*58:01 allele with allopurinol-induced SCAR and the role of HLA-B*58:01 testing to prevent allopurinol-induced SCAR were included in the participant information sheet, The participants were allowed to clarify any queries on HLA-B*58:01 testing with the researchers at the beginning of the interviews.

In-depth interviews (IDIs) were conducted for patients, while focus group discussions (FGDs) and in-depth interviews were used for the doctors. FGDs were used in the initial phase of the research, where doctors working in the same setting (junior doctors in UMMC and health clinics) were grouped together for interviews to generate broader discussion. Subsequent interviews with doctors were conducted via IDIs to allow more in-depth discussion regarding issues or ideas raised during the FGDs. Interviews were conducted by either WLN, YKL and TFAM. WLN and TFAM are clinicians and medical lecturers in the Department of Primary Care Medicine, UMMC. YKL is a senior lecturer in the same department with a background in psychology. Interviews were conducted in Malay or English, depending on the participants' preferred language as all researchers were fluent in both languages. Eight doctor participants were colleagues of the interviewers. The patient participants were not known to the interviewers prior to the study. Regarding the HLA-B*58:01 testing, the available testing modalities differed based on the setting. In UMMC, HLA-B*58:01 testing is readily available. In health clinics and private GP clinics, the blood samples had to be sent to remote private laboratories or the nearest tertiary public hospital.

All FGDs and some IDIs were conducted face-to-face. However, most of the other interviews were conducted using the online platform Zoom (Zoom Video Communications, California) due to the COVID-19 pandemic. To ensure confidentiality, all teleconference sessions were only accessible using a password. Participants were allowed to turn their video off if they were uncomfortable keeping it on. After obtaining consent, the online interviews were recorded using Zoom's function while the face-to-face interviews were audio recorded with a digital recorder. No one else besides the participant and researcher (WLN, YKL or TFAM) was present during the interview. Researchers took notes during the interviews. The IDIs took approximately 30 minutes while the FGDs took 50–60 minutes.

## Data analysis

We stopped recruitment after reaching data saturation when no additional responses were found from patients' and physicians' interviews, and no new themes emerged. One to two additional interviews were then conducted with patients and doctors to confirm data

saturation further. Descriptive statistics were used to describe the participants' characteristics. The audio recordings were transcribed verbatim and anonymised. Two researchers (WLN and NH) analysed the transcripts independently using the thematic approach. Like WLN, NH is a clinician and medical lecturer in the Department of Primary Care Medicine, UMMC. First, they read and re-read the transcripts to familiarise themselves with the content, followed by open and axial coding. NVivo software was used to manage the data. The findings between the doctors and patients were compared to identify similarities or differences in their opinions and contextualise the doctors' responses to the patients' situation. The researchers met to compare and discuss the coding and reconcile discrepancies via a team discussion. The researchers critically examined and reflected on their roles throughout the study to reduce potential biases during the interviews and data analysis. Participants provided feedback on the findings. Transcripts were analysed in either Malay or English as all researchers were familiar with these languages; participant quotes in Malay were translated into English for reporting.

## Results

We interviewed 18 patients via IDIs and 18 doctors via FGDs and IDIs (two FGDs consisting of six doctors in the first session, three doctors in the second session, and nine doctors via IDIs). Six patients refused to participate after we contacted them, but we did not pursue the reasons for refusal. None of the doctors we approached refused participation.

The characteristics of our sample are shown in Table 1. All our patients received follow-up for gout in different health settings, namely the university hospital, health clinics and private

**Table 1. Demographics and characteristics of the participants.**

| Patients (N = 18) | |
|---|---|
| **Parameters** | **Description** |
| Age | 27–75 years old (median age: 42, interquartile range: 21) |
| Gender | 16 males, 2 females |
| Ethnicity | 10 Chinese, 8 Malays |
| Education level | 5 secondary school, 3 diploma, 10 degrees |
| Duration of gout | 1–33 years (median duration: 5.5 years, interquartile range: 9) |
| On allopurinol or not | 12 patients were currently on allopurinol. |
| | 5 patients were never started on allopurinol. |
| | One patient used allopurinol before, but allopurinol was withheld due to deteriorating renal function. |
| Healthcare setting | 9 from university hospital, 5 from private general practitioner clinics, 4 from health clinics |
| **Doctors (N = 18)** | |
| **Parameters** | **Description** |
| Age | 30–44 years (mean 36, standard deviation 3.7) |
| Gender | 13 females, 5 males |
| Ethnicity | 10 Chinese, 5 Malays, 3 Indian |
| Years of practice | 3–14 years (mean 8, standard deviation 2.5) |
| Postgraduate qualification | 9 with postgraduate qualifications in primary care medicine |
| | 6 postgraduate trainees in primary care medicine |
| | 3 medical officers without postgraduate training |
| Health setting | 8 in university hospital |
| | 6 in health clinics |
| | 4 private general practitioners |

Four themes emerged from our interviews with the patients and doctors.

GP clinics. Our patients were a mixture of allopurinol users, non-users and one ex-user. None of our patients had history of severe cutaneous adverse reaction (SCAR). Only one patient had a history of rashes during initiation of allopurinol, which resolved subsequently, with the patient continuing the allopurinol. All doctors in our interview were primary care doctors with experience managing gout.

## Theme 1: HLA-B*58:01 testing can potentially improve counselling in allopurinol initiation

In our study, we discovered some practice gaps concerning counselling for allopurinol initiation, where some of our doctor participants did not elaborate on the side effects of allopurinol on the patients. The doctors were aware of the cutaneous side effects of allopurinol. They would disclose the possibility of cutaneous reactions requiring hospitalization but avoid mentioning the possibility of life-threatening reactions. There were concerns that patients would hesitate to take the medication if they learnt of the risk of severe skin reactions, like SCAR. Some doctors would emphasize that the SCAR was rare to reassure the patients. If the patient could not take allopurinol for various reasons (e.g. minor rashes, not tolerating medicine), the doctors resorted to lifestyle modification and other medication adjustments to control chronic gout instead of considering alternative urate-lowering therapy.

> "Yes, but I won't describe, I won't elaborate a lot on it. Basically, to tell them that there will be a severe rash that might cause them to be hospitalized." (Doctor 15, female, 12 years of experience, private practice, had postgraduate qualification)

> "*I don't, to be honest, I don't really go into detail about how severe the skin allergy can be."* (Doctor 12, female, 9 years of experience, private practice)

> "*Oh, there're patients, when they learnt about side effect like this, will say I'm scared, I don't want to take medication, I will try to control diet. Some of them refused to take medicine."* (Doctor 17, female, 10 years of experience, health clinic)

> "*I have to say the word it's rare, actually it's more reassuring."* (Doctor 15, female, 12 years of experience, private practice, had postgraduate qualification)

> "*In our practice nowadays, we don't have the HLA test, so what we do is if I see patient concomitantly had hypertension on perindopril, then I will switch to losartan; losartan has uricosuric effect that might help to reduce the uric acid little bit."* (when asked how she dealt with patients who cannot tolerate allopurinol) (Doctor 17, female, 10 years of experience, health clinic)

Interviews with our patients corroborated this finding. Some patients were not informed of side effects of allopurinol during initiation and were unaware of allopurinol-induced SCAR. Patients were also unaware of the existence of alternative urate-lowering therapy besides allopurinol, as their doctors did not offer them any other alternative if they could not tolerate allopurinol.

> "*No. None of the doctors ever mentioned this (when asked about side effects of allopurinol)."* (Patient 6, 39-year-old Malay male, had gout for 2 years, on allopurinol, follow-up in university hospital)

> "*So I asked the doctor, what other medicine I can take besides allopurinol (following an episode of worsening knee swelling post allopurinol). Then he said no, this is only the medicine*

*available for gout treatment, so then I said no choice." (Patient 7, 51-year-old Chinese male, had gout for 20 years, on allopurinol, follow-up in health clinic)*

In line with concerns expressed by our doctors, some of our patients who were already on allopurinol indicated awareness of rare adverse side effects may lead to non-adherence. One patient commented that he might still continue with allopurinol but would take it in as-needed manner (only when he had acute gout attack).

"*Most likely a definite no." (Patient 17, 35-year-old Chinese male, had gout for 6 years, on allopurinol, follow-up in health clinic)*

"*When I see the risk such as allergy and all, I'll be afraid to take. I may take but with little amount or rarely so. I'll take when necessary only." (Patient 6, 39-year-old Malay male, had gout for 2 years, on allopurinol, follow-up in university hospital)*

For doctors, the availability of this test would help facilitate their counselling on side effects when initiating allopurinol as they would have a more objective way to stratify if their patients were at higher risk of SCAR. The test would allow doctors to disclose the side effects more easily as they would have the option to perform this test to allay patient's concern if present. It would be a step forward from the usual trial-and-error method, which would discourage some patients from taking allopurinol. Similarly, our patients concurred that the test result would make it easier for them to decide whether to take allopurinol or not after learning the potential side effects.

"*Yeah, yeah, definitely useful, HLA testing because if there is a definitive objective result to prove that you are not at high risk in getting this, Steven Johnson Syndrome, it will be more convincing for such patient to start allopurinol." (Doctor 17, female, 10 years of experience, health clinic)*

"*The discussion on side effects will be easier, especially for those who are hesitant to disclose too many details on side effects. With the testing, you can allay the patient's concern better" (Doctor 18, female, 7 years of experience, university hospital)*

"*It is actually to help you filter the risk factor. So if the patient has the opportunity to do conduct this gene test, it makes it easier for him to make decision whether you can start this long-term medicine treatment." (Patient 16, 34-year-old Chinese male, had gout for 4 years, on allopurinol, follow-up in private GP clinic)*

### Theme 2: Decision to do HLA-B*58:01 testing in primary care

Generally, doctors and patients were receptive to implementing HLA-B*58:01 testing in primary care settings although they were not familiar with the test. Safety of medicine and prevention of severe side effects were the main positive factors linked with acceptability to the test.

"*Yeah, because it's very important, we don't know which patient we start allopurinol, which person can develop allergy reactions, like Steven Johnson syndrome" (Doctor 17, female, 10 years of experience, health clinic)*

"*I think it's great because it's for prevention right" (Patient 17, 35-year-old Chinese male, had gout for 6 years, on allopurinol, follow-up in health clinic)*

In deciding to do this HLA-B*58:01 testing, the patients' trust in their primary care doctor was important. While some patients would like to know more before deciding to do the testing, most would follow their doctors' advice without hesitation.

"*If it (decision to test) is from Dr. X (pseudonym), I trust him. He's a very good GP. Help me a lot, give me a lot of advice and consultation. I have good trust in him.*" (Patient 15, 68-year-old Chinese male, had gout for 3 years, on allopurinol, follow-up in private GP clinic)

"*Normally advice from your doctor, if the doctor says I need to do a genetic test, I just comply. I'm not the one that would challenge why should I do this, why should I do that?*" (Patient 11, 52-year-old Malay male, had gout for 12 years, not on allopurinol, follow-up in private GP clinic)

On the other hand, our doctors would generally encourage shared decision-making to decide whether the patient should proceed to do the testing, instead of asking patients to trust their recommendation (doctor-centred approach). They would provide the necessary information about the test, its advantages and disadvantages, and agree on whether to proceed with the test.

"*Then, of course, patients need to be informed, so I need to get enough information for me to convey to the patient so that we can have a mutual agreement of whether these tests need to be done or not.*" (Doctor 15, female, 12 years of experience, private practice)

"*We will allow the patient to decide. Give them all the information about the test. And then tell them the pros and cons and ask them to decide if they want us to help them to decide (smiles), you know, we will try to, again, like this, you know, give them the benefits of the test. What are the benefits of doing this test and disadvantages, try to help them to make a decision on it.*" (Doctor 9, female, 10 years of experience, health clinic)

In the interviews, the researchers stated that the turnaround time for HLA-B*58:01 test might require up to two weeks or more, depending on the setting. Neither doctors nor patients considered waiting time and place of testing as primary considerations to decide whether to test or not, as long as the acute symptoms were managed.

"*Let's say in the scenario of leg pain where I cannot walk, the thing is that as a doctor, what can he do immediately to alleviate my symptoms and so as long as basically my symptoms are managed, then I think is okay to wait.*" (Patient 11, 52-year-old Malay male, had gout for 12 years, not on allopurinol, follow -up in private GP clinic)

"*I think for things like that we don't have to really rush it. As long as there is result, I think it should be fine.*" (Patient 10, 38-year-old Chinese male, had gout for 6 years, not on allopurinol, follow-up in university hospital)

"*Because allopurinol is a preventative measure, it is not like the treatment. So actually, we don't really have to start it on the same day that we see the patient. So I think delay of a few weeks is acceptable.*" (Doctor 2, female, 7 years of experience, university hospital)

There was a divergent opinion regarding the decision to offer the test to patient where one doctor would choose to omit the testing if he felt there was an urgent need to start allopurinol. He thought doctors could be allowed to treat first and omit the test based on their judgement.

"*It's also case-to-case basis whereby you know if the patient is having frequent gout attack and then you will say okay, I need a sooner control. But, having said that, always you have other choices. We treat the patients first and then we see how, still go back to your trial and error. So yeah, I guess this is different from, you know, test like dengue, you need to know there and then. For this one we have quite a bit of leeway I think*" (Doctor 1, male, 6 years of experience, university hospital)

## Theme 3: Barriers in implementation of HLA-B*58:01 testing

We identified some barriers to implementing HLA-B*58:01 testing in primary care setting. Cost of testing was one of the major barriers identified, as HLA-B*58:01 test was not a subsidised test in the public healthcare setting. Cost of testing was also highlighted as the main information provided in doctors' shared decision-making, as many felt that the out-of-pocket payment would turn patients away from testing. Our doctors also questioned the cost-effectiveness of this testing as they considered the incidence of allopurinol-induced SCAR to be low.

I think the first thing first is we already take a lot of effort to, you know, to advise people to start on this allopurinol, it's not something very easy to ask people to start this long-term medication. Secondly, if we successfully convince people to start allopurinol and then I need to tell you now, there's a lot of side effects. In order to prevent you getting these side effects, I want you to do this testing. And so it will be a burden in term of financial burden to the patient as well. So if we routinely start the patient on this genetic testing, this will actually become a challenge to the patients where they may not want to start on this allopurinol anymore. (Doctor 14, male, 8 years of experience, private practice)

"*Yeah if it's available. I mean for a private practice. Obviously, the main limitation is just financial. Of course, as doctors, we can always recommend, and then it's up to the patient, whether they want to spend*" (Doctor 11, female, 9 years of experience, private practice)

"*If it's affordable and made available, I will do the test on my patients before I start allopurinol, but if it's very costly, the shared decision making has to come in place. We have to discuss with them, if they think they can afford to.*" (Doctor 16, female, 14 years of experience, university hospital)

"*But really, really very rare I think (incidence of allopurinol-induced SCAR). Maybe it's not so cost-effective per se if the test is very expensive as machine.*" (Doctor 13, male, 9 years of experience, health clinic)

Another additional cost highlighted would be the cost of second-line urate-lowering treatment (ULT). If patient tested positive for HLA-B*58:01, the doctors would have to consider second-line ULT such as febuxostat. Febuxostat was still costly (approximately RM280/USD60 for one month's supply) and might also require out-of-pocket expenses. The cost of febuxostat would be incremental as patients need to be on ULT for long-term, if not lifelong, as opposed to the cost of HLA testing which was just one-off.

"*I think it depends on my patients, if they are the type that who are more affluent, good health literacy. And they can afford the febuxostat in the future. I would be more willing to offer those types of patients the HLA test for them. But if my patient is from the poorer social economic group and they keep on suffering from gout attack, they need to be on preventive*

*medication. I think those are the patients that I would just counsel, please watch out for the side effect, anything happens come back immediately." (Doctor 2, female, 7 years of experience, university hospital)*

For the patients, the cost of testing was not as significant a discouraging factor as our doctors thought. Our patients were willing to pay for the testing if they saw the value in the testing as safety of medicine was a priority. Nevertheless, it was recognized that everyone had different spending power and thus, the actual cost would determine whether they could afford it.

*"If I see the value of it, I would do it. Because the investment is for your health, and so I will do it. Normally I can claim because I'm still working with a company now. So to my best ability, I will claim. But if I cannot claim, what to do, you need to pay myself. Well, to answer your question, in short, if I see benefit in it, because there's an appeal, just find money to afford it." (Patient 11, 52-year-old Malay male, had gout for 12 years, not on allopurinol, follow-up in private GP clinic)*

*"Because I need the medicine, I am willing to pay" (Patient 6, 39-year-old Malay male, had gout for 2 years, on allopurinol, follow-up in university hospital)*

HLA-B*58:01 testing was also seen as a potential burden on the consultation by patients; an additional step to what was already a difficult consultation to initiate allopurinol. In addition, repeated visits would be needed as the results would take an estimated two to four weeks to arrive. The doctors were concerned that patients would be discouraged from starting allopurinol due to the complicated or additional procedure to do this genetic testing. There were concerns that patients might find it tedious and default the follow-up.

*"So they may add this challenge to the patients that he feels that, you see, you asked me to start allopurinol, I wanted to start already. And now you asked me to do this testing. It may be a very complicated thing to them." (Doctor 14, male, 8 years of experience, private practice)*

Some doctors also highlighted the unavailability of alternative treatment in primary care and raised concern on referral to specialist clinic for alternative medication. Second-line ULT is only available in tertiary centres and specialist clinics such as rheumatology clinics. There is no established pathway to refer patients with positive HLA-B*58:01 to the rheumatologist, as HLA-B*58:01 testing is not a part of routine public healthcare screening policy. The doctors were also concerned the rheumatologist might not be happy to receive these patients as an additional workload.

*"Probably when we identify a lot of people allergic to allopurinol, we don't have anything else in the clinic except that we might need to report all those patients to tertiary hospital, and they might have a lot of patient load in the end." (Doctor 4, female, 7 years of experience, university hospital)*

*"Let's say there are many of them (patients with positive result for HLA-B*58:01), and it might be increasing the workload for the specialist. So not sure whether they would be happy to do that" (Doctor 18, female, 7 years of experience, university hospital)*

One patient raised a concern about data security of his genetic information. This issue was not mentioned by any doctor in our interviews. Patient 9 was concerned that his genetic information might be misused for reasons unknown to him. He was also worried that any misuse of that information may affect his future generation since we were handling genetic data.

*"The only thing I would like to mention is what about the security of the information? What they are gonna do with it? If it's secure, then it's okay because we have children you know. We don't know what's the effect of the insecurity when it comes to this type of knowledge, information."* (Patient 9, 68-year-old Chinese male, had gout for 25 years, not on allopurinol, follow-up in university hospital)

In our interviews, none of the participants raised concern about discrimination by insurance companies if they obtained the genetic data.

### Theme 4: Preference for targeted screening

Most of our doctors indicated that they favoured targeted screening instead of universal screening when it came to HLA-B*58:01 screening prior to allopurinol initiation. The preference for targeted screening was driven mainly by the cost-effectiveness of the screening. Ethnicity, particularly the Chinese, and history of allergies were pointed out as potential risk factors for screening. However, some doctors would offer the test to all patients with gout, provided it was cost-effective and the results were valid and fast.

*"I suppose if there are proper guidelines and criteria on who should take the HLA test, then I think that will be better. So you know, we don't need to ask everybody to do it, just those who actually have the risk of getting the side effects."* (Doctor 11, female, 9 years of experience, private practice)

*"I think it should be like, if you're talking about screening, it should be a target group. Of course, it is good that we can screen everyone. But when you want to do screening, it is always about the cost-effectiveness, so, of course, if cost is not an issue, we can screen everyone that we want to be on allopurinol. If cost is an issue, maybe we need to be certain they have fulfilled certain criteria, or maybe like if they are Chinese, you know, then they're higher risk, you know, or like some other risk factor, that they have higher risk, then maybe we should put them to our priority list to the screening. That's what I think I think all the disease should be the same. I think it's just like gestational diabetes, we have a set of criteria, this factor to be screened."* (Doctor 13, male, 9 years of experience, health clinic)

*"So, as far as I know the prevalence is higher among Chinese and also for Asian type if I'm not mistaken. And because of that, we need to balance between the cost. I don't have knowledge about the cost. So if the cost is not very high, then perhaps we can implement that. Not as a universal testing to everyone but more to targeted patients, group of patients yeah."* (Doctor 16, female, 14 years of experience, university hospital)

*"If result is valid, is fast and cost effective, I think I'm willing to offer to all my patients."* (Doctor 14, male, 8 years of experience, private practice)

*"If we are implementing such a test, and I will test for every patient that I think needs allopurinol."* (Doctor 10, male, 7 years of experience, health clinic)

One of the patients also concurred with the notion of selective screening. She mentioned that every patient should have a personalized risk assessment to decide whether they needed testing.

*"Other than that indication, we have to decide how serious it is. That means individuals know their own usual self, right? Every patient is different, you cannot base on the doctors' knowledge. It is general, you know what I mean, general ways, general compilation of all these*

*research. So every person is different." (Patient 14, 62-year-old Chinese female, had gout for 3 years, not on allopurinol, follow-up in university hospital)*

## Discussion

There are several principal findings from our study. First, we discovered inadequate disclosure of severe side effects of allopurinol by doctors due to concerns about medication refusal by patients, which could potentially be improved by introducing HLA-B*58:01 testing. Second, regarding the decision to test for HLA-B*58:01, patients tended to rely on doctors' advice, although our doctors preferred the shared decision approach. Both doctor and patient participants preferred targeted screening for HLA-B*58:01 based on risk factors such as ethnicity. However, there are barriers to the implementation of HLA-B*58:01 testing, such as out-of-pocket cost for patients, the cost-effectiveness of this implementation, lack of established alternative treatment pathway besides allopurinol, counselling burden and concern about genetic data security.

When we explored the current practice of allopurinol initiation, we identified inadequate disclosure of the severe side effects of allopurinol by the doctors out of concern that it would discourage patients from taking allopurinol. This situation is not unique to allopurinol. Tarn et al. (2006) demonstrated that doctors frequently neglected to communicate important aspects of medication, such as side effects when introducing new medicines [22]. This can potentially lead to medicolegal issues if patients developed SCAR from allopurinol but were not adequately informed about it. While some of our patients stated that they would have avoided allopurinol if they had known the full extent of the side effects such as SCAR, this concern can be alleviated by improving their understanding of incidence of allopurinol-induced SCAR and the "start low and slow" approach [23]. Studies have shown that patients who received more information from their doctors about medications and their adverse effects had higher adherence to treatment [24–26]. This is where HLA-B*58:01 testing can fill the gap in counselling for allopurinol initiation, where the test can provide a more definitive measure of risk to get SCAR, as alluded by our doctors. A shared decision approach can be employed where patient can make a more informed decision based on the HLA testing along with other risk factors, rather than counting on the "trial and error" approach [27].

We also noticed that there was hardly any discussion by doctors on the alternative ULT for gout besides allopurinol. It is understandable as alternative ULT were not readily available in public and private primary care setting in Malaysia. Febuxostat is available in the tertiary setting but the cost of treatment is another consideration, either at the expense of subsidy or out-of-pocket cost, as febuxostat is still quite costly. This may lead to allopurinol seemingly being the only treatment available for chronic gout. As revealed by some of our patients in the interview, they felt obliged to take it despite the concern of side effects as they assumed there was no other alternative. In the practical sense, the revelation of alternative treatment would not change the course of management as allopurinol is still the first line. However, the availability of HLA testing will provide an additional assurance to patients that they have a lower risk of SCAR with allopurinol if tested negative, hence improving the uptake. We also hypothesized that the introduction of HLA-B*58:01 testing would stimulate discussion on alternative ULT as part of the treatment pathway for patients who tested positive for HLA-B*58:01. This may help reduce over-reliance on lifestyle modification to control chronic gout. Lifestyle modifications alone, such as dietary control and weight loss, have no strong evidence of benefit for chronic gout [28, 29]. Overdependence on lifestyle modification alone to treat chronic gout may contribute to the underutilisation of ULT and undertreatment. This highlights the need

for a clearer pathway to initiate alternative ULT via referral to tertiary centres or improve accessibility and awareness in primary care.

Regarding the acceptability to the test, generally the doctors and patients were receptive to the test although they were unfamiliar with it. They were prepared to implement the test provided that adequate training is given. This contrasts with the findings from the literature where primary care physicians frequently felt unprepared or uncomfortable to apply pharmacogenomic testing [30, 31]. However, the general acceptability of the test in our study could signal that doctors and patients are naïve when it comes to genomic testing in terms of the procedures needed and understanding of risk in genetic counselling. They may view it as just another routine blood test for chronic diseases. Our patients trust their doctors to decide whether to do the test. While it is common to assume that Asians, particularly Southeast Asians, prefer a more doctor-centred consultation, the current evidence suggests that the paradigm is shifting towards a shared decision or partnership model in healthcare consultation [32, 33]. The lack of health literacy and understanding of the test may explain the lack of patient involvement in decision-making to do the test [34]. Nevertheless, this points out that the doctor's counselling and encouragement to do the test will play an important role in the uptake of HLA-B*58:01 testing. A shared decision approach, as suggested by most of our doctor participants, would be a more appropriate approach for HLA-B*58:01 testing, as there are other considerations for the patient, such as the cost of testing, the cost of alternative treatment and the anxiety associated with a positive test.

The waiting time for the test is not a major concern for our doctors and patients. The decision to start allopurinol is not urgent, and in current practice, it is usually initiated two weeks after the acute event if the patient is indicated for ULT. Hence, the interval may serve well as a waiting time for test result first, with doctor initiating the discussion during the acute attack, with patient having more time to consider their decision. Although there is evidence to show allopurinol can be started during the acute phase (which is useful for patients who are likely to default), it is not commonly practised. Our interviews also showed that our patients were willing to return to review the test result. Hence, the patient not returning for follow-up visits may not be a significant concern. Point-of-care testing for HLA-B*58:01 is an interesting notion that may resolve some of the concerns with patient needing repeated visits for the test. However, the accuracy of such test needs to be evaluated vigorously before implementation.

The cost of testing was singled out as the important barrier to testing by the doctors. As mentioned earlier, HLA-B*58:01 is not a subsidised test in a nationally funded health service. While patients were agreeable to bear the cost if it was within their economic capabilities, the variation in each individual's financial standing will make it challenging to implement HLA-B*58:01 testing on a broader scale, as it may not be within their means, hence exacerbating health inequalities. To convince public healthcare to absorb the cost of testing, the cost-effectiveness data is important, as pointed out by our doctors. While studies in Thailand and Korea demonstrated that HLA-B*58:01 testing was cost-effective, cost-effectiveness studies in Malaysia and Singapore showed otherwise [35–38]. However, some researchers suggested a future recalculation of the cost-effectiveness of pre-treatment HLA-B*58:01 screening based on updated local data on HLA-B* 58:01 allele frequencies for multiple ethnic groups, effect size for association of this allele with allopurinol-induced SCAR and incidence rate of allopurinol-induced SCAR in Malaysia [10]. It may be more cost-effective to do targeted screening instead of universal testing for patients with gout.

The notion of targeted screening was proposed by our doctors, where they singled out those of Chinese ethnicity. Previous studies on the prevalence and incidence of allopurinol-induced SCAR in Malaysia support this proposal, but the prevalence and incidence were high among the individuals of Malay ethnicity as well, which is the majority ethnic group in

Malaysia [7, 39, 40]. Hence, it would be prudent to consider extending screening to individuals other than those of Chinese ethnicity in Malaysia. Other clinical risk factors associated with higher risk of allopurinol-induced SCAR include renal impairment, older age, female gender, and the use of diuretics [8, 41–43]. American College of Rheumatology only suggests selective screening in Southeast Asian populations based on ethnicity such as the Han Chinese, Korean, Thai. However, this recommendation may not be adequate for a cost-effective targeted approach. A more precise recommendation with a clinical decision support system to assess who is at higher risk, incorporating various risk factors such as ethnicity and clinical factors as mentioned above, is ideal. Future research is needed to evaluate this potential decision support system.

### Strength and limitations

We explored this topic from both the perspectives of doctors and patients, in different settings (health clinics, university hospital and private general practitioner clinics) and via purposeful sampling on diverse demographics and clinical profiles, thus allowing us to compare the issues raised in a wider scope. Our interviewers are experienced researchers in qualitative methods. The interviewers carefully reviewed their roles during the study and obtained respondent validation to minimize biases in interviews and data analysis.

There were several considerations that may affect the interpretation of our findings. Firstly, most of our participants were not familiar with the test. The information on HLA-B*58:01 testing provided before the interviews did not include specific characteristics of the test such as the baseline risk, sensitivity and specificity of test, positive and negative predictive values and number needed to treat to prevent one allopurinol-induced SCAR. Hence their opinions and perceived value of the test may be based on their own supposition, which were not captured adequately in the study. Second, our findings may not be generalizable to all populations due to several factors. Our study was carried out in several health centres and clinics in Klang Valley, an urban area in Malaysia where health services are readily accessible to the population and the health literacy of our subjects may be higher. The issues raised in this study may differ from the participants in a rural setting. The ethnic diversity in our samples is limited by the absence of any Indian patients, the third largest ethnic group in Malaysia. We did not interview any patients with experience of allopurinol-induced SCAR, although it is likely that these patients would favour HLA-B*58:01 testing.

### Conclusion

There was inadequate disclosure of severe side effects of allopurinol by doctors, which could potentially be improved by discussion on HLA-B*58:01 testing. Our patients tended to rely on doctors' advice to decide on the testing although the doctors preferred the shared decision approach. Targeted screening for HLA-B*58:01 based on risk factors was the preferred option. Acceptability to the test was good although there were implementation barriers to be overcome.

Implementation of HLA-B*58:01 testing in primary care setting is potentially feasible if these three main issues are addressed: (1) a targeted screening with shared decision-making approach is adopted, (2) a cost-effective model of targeted screening approach can be developed, and (3) a clear alternative treatment pathway is made available for those tested positive.

### Supporting information

**S1 File. Interview guide for doctors.**
(DOCX)

**S2 File. Interview guide for patients.**
(DOCX)

## Acknowledgments

We would like to thank the Director General of Health Malaysia for his permission to publish this article. We also appreciate the assistance of the following people in helping us with the recruitment: Madam Noor Asyikkin binti Omar and Madam Rozlaily binti Badrulzaman from Ministry of Health Malaysia.

## Author Contributions

**Conceptualization:** Wei Leik Ng, Norita Hussein, Chirk Jenn Ng, Nadeem Qureshi, Yew Kong Lee, Zhenli Kwan, Boon Pin Kee, Sue-Mian Then, Tun Firzara Abdul Malik, Fatimah Zahrah Mohd Zaidan, Siti Umi Fairuz Azmi.

**Data curation:** Wei Leik Ng, Norita Hussein, Chirk Jenn Ng.

**Formal analysis:** Wei Leik Ng, Norita Hussein, Chirk Jenn Ng, Yew Kong Lee.

**Funding acquisition:** Wei Leik Ng, Norita Hussein, Chirk Jenn Ng, Yew Kong Lee, Zhenli Kwan, Boon Pin Kee, Sue-Mian Then, Tun Firzara Abdul Malik.

**Investigation:** Wei Leik Ng, Norita Hussein, Yew Kong Lee, Tun Firzara Abdul Malik, Fatimah Zahrah Mohd Zaidan, Siti Umi Fairuz Azmi.

**Methodology:** Wei Leik Ng, Norita Hussein, Chirk Jenn Ng, Nadeem Qureshi, Yew Kong Lee, Zhenli Kwan, Boon Pin Kee, Sue-Mian Then, Tun Firzara Abdul Malik, Fatimah Zahrah Mohd Zaidan, Siti Umi Fairuz Azmi.

**Project administration:** Wei Leik Ng, Zhenli Kwan.

**Resources:** Wei Leik Ng, Boon Pin Kee, Fatimah Zahrah Mohd Zaidan, Siti Umi Fairuz Azmi.

**Software:** Wei Leik Ng, Yew Kong Lee.

**Supervision:** Wei Leik Ng, Norita Hussein, Chirk Jenn Ng, Nadeem Qureshi.

**Validation:** Wei Leik Ng, Norita Hussein.

**Visualization:** Wei Leik Ng, Norita Hussein, Zhenli Kwan, Boon Pin Kee.

**Writing – original draft:** Wei Leik Ng.

**Writing – review & editing:** Wei Leik Ng, Norita Hussein, Chirk Jenn Ng, Nadeem Qureshi, Yew Kong Lee, Zhenli Kwan, Boon Pin Kee, Sue-Mian Then, Tun Firzara Abdul Malik, Fatimah Zahrah Mohd Zaidan, Siti Umi Fairuz Azmi.

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
