## [Decision Letter · Decision Letter 0]

23 Oct 2023

PONE-D-23-27384Implementing HLA-B*58:01 testing prior to allopurinol initiation in Malaysian primary care setting: A qualitative study from doctors' and patients' perspectivePLOS ONE

Dear Dr. Ng,

Thank you for submitting your manuscript to PLOS ONE. After careful consideration, we feel that it has merit but does not fully meet PLOS ONE’s publication criteria as it currently stands. Therefore, we invite you to submit a revised version of the manuscript that addresses the points raised during the review process.

Please submit your revised manuscript by Dec 07 2023 11:59PM.  If you will need more time than this to complete your revisions, please reply to this message or contact the journal office at plosone@plos.org. Please include the following items when submitting your revised manuscript:A rebuttal letter that responds to each point raised by the academic editor and reviewer(s). You should upload this letter as a separate file labeled 'Response to Reviewers'.A marked-up copy of your manuscript that highlights changes made to the original version. You should upload this as a separate file labeled 'Revised Manuscript with Track Changes'.An unmarked version of your revised paper without tracked changes. You should upload this as a separate file labeled 'Manuscript'.If applicable, we recommend that you deposit your laboratory protocols in protocols.io to enhance the reproducibility of your results. Protocols.io assigns your protocol its own identifier (DOI) so that it can be cited independently in the future. For instructions see: https://journals.plos.org/plosone/s/submission-guidelines#loc-laboratory-protocols. Additionally, PLOS ONE offers an option for publishing peer-reviewed Lab Protocol articles, which describe protocols hosted on protocols.io. Read more information on sharing protocols at https://plos.org/protocols?utm_medium=editorial-email&utm_source=authorletters&utm_campaign=protocols.

We look forward to receiving your revised manuscript.

Kind regards,

Chim C. Lang

Academic Editor

PLOS ONE

Journal Requirements:

Additional Editor Comments :

Please attend to reviewer's comments

Reviewers' comments:

Reviewer's Responses to Questions

**Comments to the Author**

1. Is the manuscript technically sound, and do the data support the conclusions?

Reviewer #1: Partly

2. Has the statistical analysis been performed appropriately and rigorously? 

Reviewer #1: N/A

3. Have the authors made all data underlying the findings in their manuscript fully available?

Reviewer #1: Yes

4. Is the manuscript presented in an intelligible fashion and written in standard English?

Reviewer #1: Yes

5. Review Comments to the Author

Reviewer #1: Thank you for giving me the opportunity to comment on this interesting and well written submission.

I have some comments that relate to the limitations of this work.

One of the major issues stems from the fact that neither the clinicians nor the patients were familiar with the test. Hence, their opinions and perceived value of such a test would not be based on direct knowledge or data, and it would simply be based on supposition and assumptions. It would have been far more helpful to give them an information sheet that briefly summarises the characteristics of the test, including baseline probability of harm, positive and negative predictive values and number needed to test to prevent one serious adverse reaction. This method would have enabled capture of opinions based on genuine fact, rather than anybody's guess, particularly if the participants may have wildly differing ideas about the predictive values of such a test. In such a situation, it would have been very helpful to capture the participants perception of the predictive values and to assess if this is associated with heterogeneity in the opinion.

It would help to know the average income in Malaysia, so that we can fully understand the context of the cost of such a test and treatments, as well as the need to pay for further clinic appointments at a private centre.

I'm surprised that none of the interviews were conducted in Chinese given that substantial numbers of the patients were of Chinese ethnicity. It seems to me therefore that the patient sample likely consisted of more highly educated people who were fluent in two or more languages. I feel that this qualitative study lacks generalizability because of this particular sample, and the lack of Indian ethnicity.

There is a sentence in the limitation section where it is claimed that the sample from the Klang Valley would not differ substantially from the rural population. I don't think that this statement is supported by evidence, and my personal experience is that rural patients have very different ideas and expectations as compared to the large urban areas.

6. PLOS authors have the option to publish the peer review history of their article (what does this mean?). If published, this will include your full peer review and any attached files.

Reviewer #1: No

---

## [Author Response · Author response to Decision Letter 0]

20 Nov 2023

Dear Editor and reviewers, 

Thank you for giving us the opportunity to submit a revised draft of our manuscript titled “Implementing HLA-B*58:01 testing prior to allopurinol initiation in Malaysian primary care setting: A qualitative study from doctors' and patients' perspective”. We appreciate the time and effort that you and the reviewer have dedicated to providing your valuable feedback on my manuscript. We have made some changes to our manuscript to address the points raised by our reviewer. We have highlighted the changes within the manuscript.

Here is a point-by-point response to the reviewer’s comments.

(the page and line number mentioned below are based on the tracked changes version of the manuscript)

Comment:

I have some comments that relate to the limitations of this work.

One of the major issues stems from the fact that neither the clinicians nor the patients were familiar with the test. Hence, their opinions and perceived value of such a test would not be based on direct knowledge or data, and it would simply be based on supposition and assumptions. It would have been far more helpful to give them an information sheet that briefly summarises the characteristics of the test, including baseline probability of harm, positive and negative predictive values and number needed to test to prevent one serious adverse reaction. This method would have enabled capture of opinions based on genuine fact, rather than anybody's guess, particularly if the participants may have wildly differing ideas about the predictive values of such a test. In such a situation, it would have been very helpful to capture the participants perception of the predictive values and to assess if this is associated with heterogeneity in the opinion.

Reply:

We acknowledge this limitation which would affect the interpretation of our findings. While we did provide information regarding HLA-B*58:01 testing to our participants prior to the interviews, we did not include detailed information such as the baseline risk and accuracy of the test. 

We have added a statement in Methods to inform regarding the provision of some basic information to our participants:

“Information regarding the risk of SCAR with allopurinol, association of HLA-B*58:01 allele with allopurinol-induced SCAR and the role of HLA-B*58:01 testing to prevent allopurinol-induced SCAR were included in the participant information sheet, The participants were allowed to clarify any queries on HLA-B*58:01 testing with the researchers at the beginning of the interviews.” (line 160-164, page 7-8)

We have also addressed this limitation in our Strength and limitations section:

“There were several considerations that may affect the interpretation of our findings. Firstly, most of our participants were not familiar with the test. The information on HLA-B*58:01 testing provided before the interviews did not include specific characteristics of the test such as the baseline risk, sensitivity and specificity of test, positive and negative predictive values and number needed to treat to prevent one allopurinol-induced SCAR, Hence their opinions and perceived value of the test may be based on their own supposition, which were not captured adequately in the study.” (line 589-595, page 26-27)

Comment:

It would help to know the average income in Malaysia, so that we can fully understand the context of the cost of such a test and treatments, as well as the need to pay for further clinic appointments at a private centre.

Reply:

We have included this information in the introduction:

“The median monthly income for employed citizens in Malaysia was RM2400 (USD 518) with an average monthly household income of RM8479 (USD 1831) in 2022.” (line 103-105, page 5)

Comment:

I'm surprised that none of the interviews were conducted in Chinese given that substantial numbers of the patients were of Chinese ethnicity. It seems to me therefore that the patient sample likely consisted of more highly educated people who were fluent in two or more languages. I feel that this qualitative study lacks generalizability because of this particular sample, and the lack of Indian ethnicity.

Reply:

The reviewer is correct in pointing this out as our Chinese patients were able to converse well in either English or Malay. While it is not unusual for the Chinese population in Malaysia to converse in two or more languages (Chinese in various dialects, English and Malay), we acknowledge that our patient sample consisted of more highly educated people. All our patient participants had at least secondary school education level with 10 of them with university degree. We took note of this and had addressed that the health literacy of our participants were likely higher in line 598, page 27. 

We had revised our sentences in Strength and Limitations section to note the limited generalizability of our findings: 

“Second, our findings may not be generalizable to all populations due to several factors. Our study was carried out in several health centres and clinics in Klang Valley, an urban area in Malaysia where health services are readily accessible to the population and the health literacy of our subjects may be higher. The issues raised in this study may differ from the participants in a rural setting. The ethnic diversity in our samples is limited by the absence of any Indian patients, the third largest ethnic group in Malaysia.” (line 595-600, page 27)

Comment:

There is a sentence in the limitation section where it is claimed that the sample from the Klang Valley would not differ substantially from the rural population. I don't think that this statement is supported by evidence, and my personal experience is that rural patients have very different ideas and expectations as compared to the large urban areas.

Reply:

We have rephrased our sentence in the Strength and Limitations section to acknowledge that our findings may differ from those in the rural population: 

“The issues raised in this study may differ from the participants in a rural setting.” (line 598-599, page 27)

In addition to the above comments, we have also made some corrections to the spelling and grammatical errors.

We look forward to hearing from you regarding our submission and to respond to any further questions and comments you may have.

---

## [Editor Report · Decision Letter 1]

15 Dec 2023

Implementing HLA-B*58:01 testing prior to allopurinol initiation in Malaysian primary care setting: A qualitative study from doctors' and patients' perspective

PONE-D-23-27384R1

We’re pleased to inform you that your manuscript has been judged scientifically suitable for publication and will be formally accepted for publication once it meets all outstanding technical requirements.

Kind regards,

Chim C. Lang

Academic Editor

PLOS ONE

Additional Editor Comments (optional):

\\This revised manuscript is much improved.
---

## [Editor Report · Acceptance letter]

3 Jan 2024

PONE-D-23-27384R1 

PLOS ONE

Dear Dr. Ng, 

I'm pleased to inform you that your manuscript has been deemed suitable for publication in PLOS ONE. Congratulations! Your manuscript is now being handed over to our production team.

Kind regards, 

on behalf of

Dr. Chim C. Lang 

Academic Editor

PLOS ONE